# Preparation of Sesquiterpene Lactone Derivatives: Cytotoxic Activity and Selectivity of Action

**DOI:** 10.3390/molecules24061113

**Published:** 2019-03-20

**Authors:** María F. Beer, Augusto E. Bivona, Andrés Sánchez Alberti, Natacha Cerny, Guillermo F. Reta, Víctor S. Martín, José M. Padrón, Emilio L. Malchiodi, Valeria P. Sülsen, Osvaldo J. Donadel

**Affiliations:** 1INTEQUI-CONICET, Facultad de Química, Bioquímica y Farmacia, Universidad Nacional de San Luis, Almirante Brown 1445, CP D5700HGC, San Luis, Argentina; mflorenciabeer@gmail.com (M.F.B.); gfreta@gmail.com (G.F.R.); 2CONICET-Universidad de Buenos Aires. Instituto de Química y Metabolismo del Fármaco (IQUIMEFA), Junín 956 2°P (1113), Buenos Aires, Argentina; 3Cátedra de Inmunología, Facultad de Farmacia y Bioquímica, Universidad de Buenos Aires, Instituto de Estudios de la Inmunidad Humoral (IDEHU), UBA-CONICET. Junín 956 4°P (1113), Buenos Aires, Argentina; augustobivona@gmail.com (A.E.B.); andres.sanchez.alberti@gmail.com (A.S.A.); emalchio@ffyb.uba.ar (E.L.M.); 4CONICET-Universidad de Buenos Aires. Instituto de Microbiología y Parasitología Médica (IMPaM), Facultad de Medicina. Paraguay 2155. Piso 13, Buenos Aires, Argentina; 5CONICET-Universidad Nacional de Luján, Instituto de Ecología y Desarrollo Sustentable (INEDES), Ruta 5 y Avenida Constitución-(6700), Luján, Argentina; natachacerny@gmail.com; 6Instituto Universitario de Bio-Orgánica Antonio González (IUBO-AG), Universidad de La Laguna, Avda. Astrofísico Francisco Sánchez 2, 38206 La Laguna, Spain; vmartin@ull.es (V.S.M.); jmpadron@ull.es (J.M.P.); 7Cátedra de Farmacognosia, Facultad de Farmacia y Bioquímica, Universidad de Buenos Aires, Junín 956 2°P (1113), Buenos Aires, Argentina

**Keywords:** sesquiterpene lactones, antiproliferative activity, Asteraceae, cumanin, helenalin, hymenin

## Abstract

Cancer is one of the most important causes of death worldwide. Solid tumors represent the great majority of cancers (>90%) and the chemotherapeutic agents used for their treatment are still characterized by variable efficacy and toxicity. Sesquiterpene lactones are a group of naturally occurring compounds that have displayed a diverse range of biological activities including cytotoxic activity. A series of oxygenated and oxy-nitrogenated derivatives (**4**–**15**) from the sesquiterpene lactones cumanin (**1**), helenalin (**2**), and hymenin (**3**) were synthesized. The silylated derivatives of helenalin, compounds **13** and **14**, were found to be the most active against tumor cell lines, with GI_50_ values ranging from 0.15 to 0.59 μM. The ditriazolyl cumanin derivative (**11**) proved to be more active and selective than cumanin in the tested breast, cervix, lung, and colon tumor cell lines. This compound was the least toxic against splenocytes (CC_50_ = 524.1 µM) and exhibited the greatest selectivity on tumor cell lines. This compound showed a GI_50_ of 2.3 µM and a SI of 227.9 on WiDr human colon tumor cell lines. Thus, compound **11** can be considered for further studies and is a candidate for the development of new antitumor agents.

## 1. Introduction

According to recent studies, 60% of newly identified chemical entities are natural products, semi-synthetic analogs, or synthetic compounds based on their pharmacophores [1]. This occurs together with the increased incidence of life-threatening diseases such as AIDS, cancer, hepatitis, etc. [2]. It is noteworthy that some semi-synthetic compounds derived from a natural product sometimes show higher bioactivity than the original natural compound. Considering the wide range of biological activities, such as antiparasitic, antiproliferative, anti-inflammatory, antiviral, antibacterial, and antifungal activity, sesquiterpene lactones have attracted scientific interest [3,4,5,6,7,8,9].

Analyzing the structure of molecules used in cancer therapy, most of them show nitrogenated functional groups like amide and carbamate moieties in their framework [10]. Recently, it a series of disubstituted 1,2,3-triazoles has been reported exhibiting potent cytotoxicity in the nanomolar range and tubulin inhibitory activity in the low micromolar range [11]. This kind of compound is considered an interesting unit in the design of anticancer drugs. Such heterocycles may act according to their dipolar character, rigidity, and ability to form hydrogen bridge bonds, or simply as connectors [12].

Synthetic organic chemists have shown great interest in 1,2,3-triazoles for developing new biologically active molecules. Triazole moieties do not occur naturally, but 1,2,3-triazole cores may form the basis of small-molecule pharmaceutical leads. Molecules containing this heterocyclic nucleus have being reported to have anti-HIV, antimicrobial, anti-allergic, antifungal, and antitumor activity [13].

In a previous work, we reported the cytotoxicity enhancement of some sesquiterpenes and iridoids when their lipophilicity was increased by adding alkyl and/or aryl-silyl functionalities on the natural framework [12].

In consideration of the abovementioned factors, we herein describe our findings aimed at the synthesis and cytotoxic evaluation of oxygenated and oxy-nitrogenated derivatives from sesquiterpene lactones (STLs) cumanin (**1**), helenalin (**2**) and hymenin (**3**) (Figure 1). It should be emphasized that these STLs can be isolated in significant quantities from the natural sources. Furthermore, these plants are widely distributed in Argentina [14,15,16].

## 2. Results and Discussion

### 2.1. Chemistry

The STLs cumanin (**1**), helenalin (**2**), and hymemin (**3**) were isolated from *Ambrosia tenuifolia* Spreng., *Gaillardia megapotamica* var. *megapotamica* Spreng., and *Parthenium hysterophorus* L., respectively (Figure 1).

In preliminary bioactivity tests, these STLs showed significant cytotoxic activity, with helenalin being the most active (**2**). Therefore, STLs have been used as starting materials for the preparation of a series of oxygenated and oxo-nitrogenous products; these modifications led to an improved activity of the obtained derivatives. In this way, derivatives **4** to **15** were prepared (Figure 2).

The presence of a hydroxyl group in the structure of the natural compound allowed for the preparation of acetylated and silylated derivatives from these STLs. In this regard, the diacetylated derivative **4** was prepared under standard conditions from structure **1**, and silylated derivatives **5** to **7** of were obtained under standard conditions and by varying the silylating agent (Scheme 1). The silylation reaction of cumanine with TBDPSiCl led to the monosilylated derivative at C3. The introduction of this voluminous group resulted in high steric hindrance. NMR, COSY, and HMBC analysis allowed for the coupling of H4 (δ = 3.45, dd, *J* = 8 Hz) with the hydrogen of the hydroxyl group of the position 3; HSQC analysis confirms the above since the signal corresponding to OH does not show C‒H coupling. HRMS-ES (*m/z*) analysis: [M + Na]^+^: Calcd. for C_31_H_40_O_4_Si_2_Na: 527.2594; found 527.2586, determines the presence of monosilylated derivative **7**.

Acetylated and silylated derivatives (**12** to **15**) were prepared from STLs **2** and **3** (Scheme 2).

Synthetic organic chemistry has encouraged great interest in 1,2,3-triazoles in the development of new biologically active molecules [12]. The triazole moiety does not occur in nature, but 1,2,3-triazole cores may form the basis of small-molecule pharmaceutical leads. Molecules containing this heterocyclic nucleus have been reported to have anti-HIV, antimicrobial, anti-allergic, antifungal, and antitumor activities [12]. One of the most popular reactions within the click chemistry paradigm is the Cu (I)-catalyzed 1,3-dipolar Hüisgen cycloaddition of alkynes and azides. This reaction proceeds with great efficiency and selectivity in aqueous media and yields triazole moieties [17]. Our first objective was to obtain alkynes **8** and **9** using STL **1** as the starting material (see Scheme 3). Compound 1 was treated with propargyl bromide and sodium hydride in THF to obtain derivatives **8** and **9**, and working conditions were optimized in order to increase the yield of both products (for the characterization of the propargyl group at C3 see the Appendix A). Subsequently, under Hüisgen conditions, **8** and **9** were converted into the corresponding mono and di-triazole derivatives **10** and **11**, respectively (Scheme 3).

### 2.2. Biological Results

#### 2.2.1. Antiproliferative Activity

In vitro antiproliferative activity was evaluated using the protocol of the National Cancer Institute (NCI) after 48 h of drug exposure using the sulforhodamine B (SRB) assay. Results, expressed as GI_50_ values, are shown in Table 1. Data revealed that oxygenated derivatives (**5**–**7**, **13**–**15**) are more active than natural products (**1**–**3**) with GI_50_ values ranging from 0.15 to 6.8 μM, in all cell lines. Helenalin silylated derivatives **13** and **14** were found to be the most active against all tested cell lines, with GI_50_ values ranging from 0.15 to 0.59 μM. We cannot discard that increasing lipophilicity may well result in an increase in cytotoxicity in vitro.

Most of the cumanin derivatives have shown higher cytotoxic activity than the natural parent compound **1**. The presence of two 1,2,3-triazole groups in derivative **11** has increased the activity values by one order of magnitude compared to cumanin (**1**). Compounds **4, 8, 9**, and **10** displayed moderate activity and similar results to **1,** with the exception of compounds **8** and **9**, which were more active than **1** against WiDr cells.

Hymenin derivative (**15**) has shown increased activity compared with natural compound **3**.

#### 2.2.2. Cytotoxicity on Primary Cell Culture Activity

The cytotoxicity of the sesquiterpene lactones and its derivatives was evaluated using mouse splenocytes. Table 2 shows the results of the cytotoxicity assay, as CC_50_ (the concentration of each compound that causes 50% cell death) and the selectivity indexes. Cumanin (**1**) was the least toxic natural compound against splenocytes (CC_50_ = 29.4 µM) compared with the other sesquiterpene lactones (**2** and **3**). Cumanin derivatives (**4**–**11**) have shown CC_50_ values and selectivity indexes higher than natural compound **1**. Compounds **4**–**9** displayed moderate selectivity, while compound **10** showed low selectivity on the tested cell lines. The incorporation of two triazole groups in cumanin (**1**) reduced cytotoxicity on normal cells and improved selectivity against tumor cell lines. Compound **11** was the least toxic against splenocytes (CC_50_ = 524.1 µM) and presented the greatest selectivity on the tested cell lines. This compound showed a GI_50_ of 2.3 and a SI of 227.9 on human colon tumor cell line WiDr.

Compounds **2** and **3,** as well as their corresponding derivatives, displayed a low selectivity of action against tumor cell lines. Similar results have been obtained for compound **12** by Lee et al. [18].

## 3. Materials and Methods

### 3.1. General

Unless otherwise stated, all solvents were purified by standard techniques. Reactions requiring anhydrous conditions were performed under an argon atmosphere. Anhydrous magnesium sulfate was used for drying solutions. Reactions were monitored by thin-layer chromatography (TLC) on silica gel plates (60 F254) (Merck KGaA, Darmstadt, IN, USA), and visualized with UV light, 2.5% phosphomolybdic acid in ethanol, or vanillin with acetic and sulfuric acid in ethanol with heating. Purification was performed by column chromatography (CC) on silica gel (230–400 mesh) using n-hexane and ethyl acetate gradient as solvent. ^1^H NMR spectra were recorded on a Bruker (Bruker Biospin GmbH, Silberstreifen, Rheinstetten, Germany) 200, 500, or 600 MHz, ^13^C NMR spectra were recorded at 50 and 125 M Hz, and chemical shifts are reported relative to internal Me_4_Si (δ = 0). Melting points were determined by using an Electrothermal IA9000 melting point apparatus; results are reported in degrees Celsius and are uncorrected. Optical rotations were recorded on a 343 Perkin Elmer polarimeter (Waltham, MA, USA). High-resolution ESI mass spectra were obtained from a Fourier transform ion cyclotron resonance (FT-ICR) mass spectrometer, a RF-only hexapole ion guide, and an external electrospray ion source. HRMS spectra were obtained on a Micromass AutoSpec (Oakville, ON, Canada) mass spectrometer.

### 3.2. Plant Material

The species used in the present work were *Ambrosia tenuifolia* Spreng. (Asteraceae), *Gaillardia megapotamica* var. *megapotamica* (Asteraceae), and *Parthenium hysterophorus* (Asteraceae).

The aerial parts of *Ambrosia tenuifolia* Spreng. (Asteraceae) were collected in May 2010 in Ibicuy, Entre Ríos Province, Argentina. The material was identified by Dr. Gustavo Giberti and the Herbarium specimen is deposited in the Museum of Pharmacobotany at the Faculty of Pharmacy and Biochemistry, University of Buenos Aires (BAF 717).

*Gaillardia megapotamica* var. *megapotamica* Spreng. (Asteraceae) was collected in November 1989 in the city of La Arenilla, San Luis Province, and identified by Prof. Eng. Luis del Vitto. The specimens are registered as Del Vito & Petenatti under No. 4633, and deposited at the Herbarium of the National University of San Luis.

*Parthenium hysterophorus* (Asteraceae) was collected in March 1988 in the town of San Roque, department of La Capital, Province of San Luis, and identified by Prof. Eng. Luis del Vitto. They are registered under the numbers 1672-UNSL, and the Herbarium specimens are deposited in the National University of San Luis.

In all cases the aerial parts were collected, dried at room temperature to a constant weight, manually fragmented or ground, as appropriate, and placed in plastic bags from which the oxygen was removed and replaced by argon inert gas for storage until the moment of processing.

### 3.3. Sesquiterpene Lactone Extraction and Isolation

#### 3.3.1. Isolation of Cumanin from *A. tenuifolia*

Extraction conditions: 1.25 kg of aerial parts dried at room temperature to constant weight were processed. The plant material was extracted with acetone (3 × 48 h) at room temperature and the organic extracts were concentrated in vacuo. The residue (30 g) was collected and chromatographed in a 63-cm, 7 cm diameter column, using Sigel 60 G (70–230 mesh) (Merck KGaA, Darmstadt, IN, USA) and eluting with n-hexane-EtOAc mixtures of increasing polarity [16].

Under these conditions, 6 g of cumanin (**1**) were obtained in the fractions eluted with hexane: ethyl acetate 2:8. These values represent a yield of 4.8 g of cumanin per kg of dry plant.

#### 3.3.2. Isolation of Helenalin from *G. megapotámica* var. *megapotámica*

The dried to constant weight aerial parts (1.50 kg) of *G. megapotamica* var *megapotamica* were extracted (3 × 48 h) with MeOH at room temperature. The organic extracts were combined and concentrated in a rotary evaporator. The dry extract was solubilized in 1 L of a mixture of MeOH: H_2_O (8:2) and stored in a refrigerator overnight. After this cooling time, the organic phase was separated and the aqueous phase was washed with MeOH (3 × 500 mL). The organic phases were combined, and solvent was removed by rotary evaporator [18]. Thirty-five grams of dry extract were recovered and chromatographed on a 65-cm length and 7 cm in diameter column, using as stationary phase of Sigel 60 G (70–230 mesh) and eluting with n-hexane:EtOAc mixtures of increasing polarity. After two separative processes in CC, 10.2 g of helenalin (**2**) were obtained, representing a yield of 6.8 g per kg of dry plant.

#### 3.3.3. Extraction and Purification of Hymenin from *P. hysterophorus*

For preparing the crude extract of the dried to constant weight aerial parts of *P. hysterophorus*, 1.10 kg of plant material was used and extracted by maceration at room temperature with CHCl_3_ for 24 h. The maceration was filtered and the extract was taken to dryness in a rotary evaporator. Subsequently, the dried extract was solubilized in hot EtOH and left overnight at room temperature. Then, it was filtered and the ethanolic solution was extracted with CHCl_3_ (4 × 150 mL). The organic phase was dried over anhydrous Na_2_SO_4_ and filtered, and the solvent was removed under reduced pressure at a temperature lower than 40 °C until 20 g of a gummy-like dry residue was obtained [18]. The fractionation was carried out by preparative CC (65 × 7 cm) using Sigel 60G (70–230 mesh) as stationary phase and eluting with mixtures of n-hexane and EtOAc of increasing polarities. After the chromatographic separation, 3.4 g of hymenin (**3**) were obtained, representing a yield of 3.1 g per kg of dry plant.

### 3.4. Chemistry

Preparation of derivative **4**: compound **1** (50 mg) (1 eq., 0.188 mmol) of **1** was dissolved in 1 mL of pyridine (0.2 M) and 1 mL of acetic anhydride (50 eq., 9.4 mmol) was added. The reaction was monitored by TLC after extraction with EtOAc in acid medium. After 6 h, the reaction was complete and quenched by pouring in saturated copper sulfate solution and extracting with ethyl acetate (3 × 50 mL each)., The organic phase was washed with distilled water (3 × 25 mL each), dried with anhydrous sodium sulfate, and concentrated under reduced pressure. Under these conditions, 49 mg of the derivative **4** was recovered as a white solid, yield 75%.

Preparation of derivative **5**: compound **1** (50 mg) (1 eq., 0.187 mmol) was dissolved in 1.5 mL of Cl_2_CH_2_ (0.1 M), 45 mg (3.5 eq., 0.658 mmol) of imidazole, and 69 mg of TMSiCl (0.08 mL, 0.657 mmol) were added; reaction was carried out at room temperature and confirmed by TLC to be complete after 3 h. The reaction was quenched with a concentrated solution of (NH_4_)_2_SO_4_ (40 mL) and extracted with Et_2_O (3 × 20 mL). The organic phase was washed with water (30 mL) and dried with Na_2_SO_4_, filtered, concentrated, and purified by isocratic CC (30 × 0.5 cm) using Sigel 60 G (70–230 mesh), as stationary phase, and a mixture of n-hexane-EtOAc (60:40) as a mobile phase to give 48 mg of compound **5** as a yellow solid, yield 63%.

Preparation of derivative **6**: compound **1** (50 mg) (1 eq., 0.187 mmol) was dissolved in 1.5 mL of Cl_2_CH_2_ (0.1 M), 45 mg (3.5 eq., 0.657 mmol) of imidazole and 87 mg of DMISOPSiCl (0.1 mL, 0.657 mmol) were added; the reaction was carried out at room temperature and confirmed by TLC to be complete after 1 h. The reaction was quenched with a concentrated solution of (NH_4_)_2_SO_4_ (40 mL) and extracted with Et_2_O (3 × 20 mL). The organic phase was washed with water (30 mL) and dried with Na_2_SO_4_, filtered, concentrated, and purified by isocratic CC (30 × 0.5 cm) using Sigel 60 G (70–230 mesh), as stationary phase, and a mixture of n-hexane:EtOAc (50:50) as mobile phase to give 51 mg of compound **6** as a yellow solid, yield 58%.

Preparation of derivative **7:** compound **1** (50 mg) (1 eq., 0.187 mmol) was dissolved in 1.5 mL of Cl_2_CH_2_ (0.1 M), 45 mg (3.5 eq., 0.657 mmol) of imidazole and 211 mg TBDPSiCl (0.2 mL, 0.657 mmol) were added; reaction was carried out at room temperature. The reaction was confirmed to be complete by TLC after 4 h. The reaction was quenched with a concentrated solution of (NH_4_)_2_SO_4_ (40 mL) and extracted with Et_2_O (3 × 20 mL). The organic phase was washed with water (30 mL) and dried with Na_2_SO_4_, filtered, concentrated, and purified by CC (30 × 0.5 cm) using Sigel 60 G (70–230 mesh), as stationary phase and 160 mL of a mixture of n-hexane:EtOAc (80:20) and 320 mL of a mixture of n-hexane:EtOAc (70:30) as mobile phase to give 43 mg of compound **7** as a yellow solid, yield 45%. 

Preparation of derivative **8**: compound **1** (100 mg) (1 eq., 0.188 mmol) was dissolved in 5 mL of dry THF (0.04 M), and 144 mg (10 eq., 3,82 mmol) of sodium hydride (60% in mineral oil) was added. The mixture was stirred for 60 min in Ar atmosphere. Subsequently, 277.6 mg (1 eq., 0.752 mmol) of TBAI and 134 mg of propargyl bromide (3 eq., 0.10 mL; 1.128 mmol) were added. The reaction was confirmed to be complete after 4 h, and was quenched with water and extracted with ethyl acetate (2 × 50 mL). The organic phases were combined, dried with Na_2_SO_4_, filtered, concentrated, and purified by CC (30 × 0.5 cm) using Sigel 60G (70–230 mesh) as stationary phase and 120 mL of CH_2_Cl_2_ and 280 mL of CH_2_Cl_2_:EtOAc (95:5) as mobile phase to give 40 mg of **8** as a white solid, yield 35%.

Preparation of derivative **9:** compound **1** (100 mg) (1 eq., 0.382 mmol) of **1** was dissolved in 5 mL of dry THF (0.04 M), 144 mg (10 eq., 3.82 mmol) of sodium hydride (60% in mineral oil) was added. The mixture was stirred for 60 min in Ar atmosphere. Subsequently, 277.6 mg (1 eq., 0.752 mmol) of TBAI and 186 mg of propargyl bromide (6 eq., 0.20 mL, 2.292 mmol) were added. The reaction was confirmed by TLC to be complete after 4 h. The reaction was quenched with water and extracted with ethyl acetate (2 × 50 mL). The organic phases were combined, dried with Na_2_SO_4_, filtered, concentrated and purified by CC (30 × 0.5cm) using Sigel 60G (70–230 mesh) as stationary phase and 120 mL of CH_2_Cl_2_ and 280 mL of CH_2_Cl_2_:EtOAc (95:5) as mobile phase to give 28 mg of **9** as a white solid, yield 21%.

Preparation of derivative **10:** compound **8** (31.7 mg) (1 eq., 0.104 mmol) of **8** was dissolved in 3 mL (0.2 M) of a 1:1 ethanol/water mixture at room temperature, 18.84 mg of benzyl azide (1 eq., 0.104 mmol),1.67 mg of CuSO_4_.5H_2_O (0.1 eq., 0.0104 mmol), and 6.18 mg sodium ascorbate (0.3 eq., 0.0312 mmol) were added. The reaction was confirmed by TLC to be complete after 96 h. The reaction was quenched with 20 mL of water and extracted with ethyl ether (3 × 30 mL). The organic phase was dried with Na_2_SO_4_, filtered, concentrated under reduced pressure, and purified by CC using Sigel 60G (70–230 mesh) as stationary phase and a mixture of n-hexane:EtOAc (50:50) as mobile phase to give 21.6 mg of **10** as a white amorphous solid. Yield 47%.

Preparation of derivative **11:** To obtain dithriazole **11**, compound **9** (20 mg) (1eq., 0.03 mmol) was dissolved in 1 mL (0.1 M) of a 1:1 ethanol/water mixture at room temperature, and 15 mg of benzyl azide was added (1 eq., 0.03 mmol), then 1 mg of CuSO_4_.5H_2_O (0.1 eq., 0.003 mmol) and 3.29 mg of sodium ascorbate (0.3 eq., 0.009 mmol) were also added. The reaction was confirmed by TLC to be complete after 96 h. The reaction was quenched with 20 mL of water and extracted with ethyl ether (3 × 30 mL). The ether phase was dried with Na_2_SO_4_, filtered, concentrated under reduced pressure. Then, it was purified by CC of Sigel 60G (70–230 mesh) using a mixture of n-hexane:EtOAc: (40:60) as mobile phase to give 8.9 mg (yield 21.9%) of compound **11** as a white amorphous solid.

Preparation of derivative **12:** 50 mg (1 eq., 0.190 mmol) of helenalin (**2**), 1 mL (50 eq., 9.5 mmol) of acetic anhydride, and 1 mL (0.2M) of pyridine were stirred at room temperature. The reaction was confirmed to be complete by TLC after 6 h. The reaction was quenched with saturated copper sulfate solution and extracted with ethyl acetate (3 × 50 mL). The organic phase was washed with water (3 × 25 mL), dried with Na_2_SO_4_, and concentrated under reduced pressure. Then, it was purified by CC of Sigel 60G (70–230 mesh) using n-hexane:EtOAc (40:60) as mobile phase. Forty-two milligrams of **12** were obtained (yield 70%) as an amorphous solid.

Preparation of derivative **13:** 50 mg (1 eq., 0.190 mmol) of helenalin (**2**), and 45 mg (3.5 eq., 0.667 mmol) of imidazole, were dissolved in 1.5 mL (0.1.M) of Cl_2_CH_2_, at room temperature. After 15 min of stirring, 0.08 mL (3.5 eq.; 0.667 mmol) of TMSiCl was added. After 6 h, the reaction was complete. The reaction was quenched with 40 mL of ammonium chloride solution and extracted with ethyl ether (3 × 20 mL). The organic phase was washed with water (30 mL) and dried with Na_2_SO_4_, filtered, concentrated, and purified in CC (30 × 0.5 cm) of Silicagel 60G (70–230 mesh) using a mixture of n-hexane:EtOAc (50:50) as eluent. Thirty-five milligrams (yield 56%) of **13** were obtained as an amorphous solid.

Preparation of derivative **14:** 50 mg (1 eq., 0.190 mmol) of **2** and 45 mg (3.5 eq., 0.667 mmol) of imidazole were dissolved in 1.5 mL (0.1M) of Cl_2_CH_2_ at room temperature. After 15 min of stirring, 0.1 mL (3.5 eq.; 0.667mmol) of DMISOPSiCl was added. After 10 h, the reaction was complete. The reaction was quenched with 40 mL of an oversaturated ammonium chloride solution and subsequently extracted with ethyl ether (3 × 20 mL). The organic phase was washed with water (30 mL) and dried with Na_2_SO_4_, filtered, concentrated, and purified in CC (30 × 0.5 cm) of Silicagel 60G (70–230 mesh) using a mixture of n-hexane:EtOAc (50:50) as eluent. 34.5 mg (yield 50%) of **14** was obtained as an amorphous solid.

Preparation of derivative **15:** 100 mg of hymenin (**3**) (1 eq., 0.351 mmol) and 84 mg (3.5 eq., 1.23 mmol) of imidazole in 1.5 mL of dichloromethane were dissolved at room temperature. After 15 min of stirring, 0.2 mL (3.5 eq., 1.23 mmol) TMSiCl was added. The reaction lasted 72 h. After this time, the reaction was poured into 40 mL of ammonium chloride solution and extracted with ethyl ether (3 × 20 mL). The organic phase was washed with water (30 mL) and dried with Na_2_SO_4_, filtered, concentrated, and purified by CC (30 × 0.5 cm) of silica gel 60G (70–230 mesh) using n-hexane:EtOAc (30:70) as eluent; 44.6 mg of **18** (yield 35%) were obtained as an amorphous solid.

### 3.5. Spectroscopic and Physical Data

Compound **1**: White solid, m.p. 122–124 °C. [α]${}_{D}^{20}$: +144.3 (*c* 11.73; CHCl_3_). IR (KBr; *υ_max_*: cm^−1^): 3412, 1745, 2908, 1275, 984. ^1^H-NMR (δ = ppm, 600M Hz): 1.41 (1H, ddd, *J* = 13 and 7 Hz, H-1); 1.66–1.73 (1H, td, *J* = 13 and 5 Hz, H-2a); 2.05–2.11 (1H, m, H-2b); 4.21 (1H, td, *J* = 8 and 5 Hz, H-3); 3.51 (1H, d, *J* = 8 Hz, H-4); 1.59 (1H, t, *J* = 15, H-6a); 1.95 (1H, dd, *J* = 4 and 15 Hz, H-6b); 3.13 (1H, m, H-7); 4.68 (1H, dtd, *J* = 12 and 5 Hz, H-8); 1.77 (1H, c, *J* = 12 Hz, H-9a); 2.13–2.18 (1H, m, H-9b); 1.97–2.03 (1H, m, H-10); 5.61 (1H, d, *J* = 3 Hz, H-13); 6.26 (1H, d, *J* = 3 Hz, H-13′); 1.04 (3H, d, *J* = 7 Hz, H-14); 0.98 (3H, s, H-15). ^13^C-NMR (δ = ppm, 150M Hz): 41.6 (C-1); 35.3 (C-2); 68.4 (C-3); 77.1 (C-4); 44.3 (C-5); 40.7 (C-6); 37.8 (C-7); 80.1 (C-8); 36.7 (C-9); 30.4 (C-10); 139.8 (C-11); 169.9 (C-12); 123.0 (C-13); 16.7 (C-14); 17.8 (C-14). HRMS-ES *m/z*: [M + Na]^+^: Calcd. for C_15_H_22_O_4_Na: 289.1416; found: 289.1422.

Compound **2**: White solid, m.p. 159–161 °C. [α]${}_{D}^{20}$: −64.56 (*c* 12.33; CHCl_3_). IR (KBr; *υ_max_*: cm^−1^): 3342, 2927, 1699, 1757, 1111, 962. ^1^H-NMR (δ = ppm, 200M Hz): 7.52 (1H, d, *J* = 6 Hz, H-2); 6.19 (1H, d, *J* = 6 Hz, H-3); 4.85 (1H, d, *J* = 9 Hz, H-6); 3.25 (1H, m, H-7); 2.00–2.10 (1H, m, H-8); 1.71–1.79 (1H, m, H-9a); 1.71–1.79 (1H, m, H-9b); 1.71–1.79 (1H, m, H-10); 5.54 (1H, d, *J* = 3 Hz, H-13); 6.26 (1H, d, *J* = 4 Hz, H-13′); 1.13 (3H, d, *J* = 6 Hz, H-14); 1.06 (3H, s, H-15). ^13^C-NMR (δ = ppm, 50M Hz): 83.2 (C-1); 164.0 (C-2); 130.6 (C-3); 208.3 (C-4); 57.5 (C-5); 79.8 (C-6); 41.8 (C-7); 25.8 (C-8); 31.7 (C-9); 37.8 (C-10); 138.9 (C-11); 170.4 (C-12); 120.8 (C-13); 17.7 (C-14); 14.8 (C-15). HRMS-ES (*m/z*): [M + Na]^+^: Calcd. for C_15_H_18_O_4_Na: 285.1103; found 285.1107.

Compound **3**: White solid, m.p. 173–174 °C. [α]${}_{D}^{20}$: −78.69 (*c* 11.17; CHCl_3_). IR (KBr; υ_max_: cm^−1^): 3450, 2972, 1741, 1282, 976. ^1^H-NMR (δ = ppm, 200M Hz): 7.52 (1H, d, *J* = 6 Hz, H-2); 6.19 (1H, d, *J* = 6 Hz, H-3); 4.85 (1H, d, *J* = 9 Hz, H-6); 3.25 (1H, m, H-7); 2.00–2.10 (1H, m, H-8); 1.71–1.79 (1H, m, H-9a); 1.71–1.79 (1H, m, H-9b); 1.71–1.79 (1H, m, H-10); 5.54 (1H, d, *J* = 3 Hz, H-13); 6.26 (1H, d, *J* = 4 Hz, H-13′); 1.13 (3H, d, *J* = 6 Hz, H-14); 1.06 (1H, s, H-15). ^13^C-NMR (δ = ppm, 50M Hz): 83.2 (C-1); 164.0 (C-2); 130.6 (C-3); 208.3 (C-4); 57.5 (C-5); 79.8 (C-6); 41.8 (C-7); 25.8 (C-8); 31.7 (C-9); 37.8 (C-10); 138.9 (C-11); 170.4 (C-12); 120.8 (C-13); 17.7 (C-14); 14.8 (C-15). HRMS-ES (*m/z*): [M + Na]^+^: Calcd. for C_15_H_18_O_4_Na: 285.1103; found 285.1107.

Compound **4**: White solid, m.p.: 85–86 °C. [α]${}_{D}^{20}$: +57.7 (*c* 4.92; CHCl_3_). IR (KBr; υ_max_: cm^−1^): 2966, 1755, 1743, 1250, 1066, 823, 598. ^1^H-NMR (δ = ppm, 200M Hz): 1.50–1.59 (1H, m, H-1); 1.71–1.91 (1H, m, H-2a); 2.10–2.27 (1H, m, H-2b); 5.25 (1H, ddd, *J* = 8 and 15 Hz, H-3); 4.78 (1H, d, *J* = 8 Hz, H-4); 1.60 (1H, s, H-6a); 1.67 (1H, m, H-6b); 3.2 (1H, m, H-7); 4.68 (1H, ddd, *J* = 8 and 16 Hz, H-8); 1.66–1.87 (1H, m, H-9a); 2.15–2.27 (1H, m, H-9b); 2.01–2.07 (1H, m, H-10); 5.56 (1H, d, *J* = 2 Hz, H-13); 6.25 (1H, d, *J* = 2 Hz, H-13′); 1.05 (3H, d, *J* = 7 Hz, H-14); 1.07(1H, s, H-15); 2.04 (3H, s, H-2′); 2.07 (3H, s, H-2″). ^13^C-NMR (δ = ppm, 50M Hz): 41.3 (C-1); 32.3 (C-2); 69.2 (C-3); 75.7 (C-4); 43.9 (C-5); 39.9 (C-6); 37.6 (C-7); 79.4 (C-8); 36.2 (C-9); 29.8 (C-10); 139.4 (C-11); 169.5 (C-12); 123.2 (C-13); 16.6 (C-14); 18.5 (C-15); 170.2 (C-1′); 20.8 (C-2′); 170.6 (C-1″); 20.6 (C-2″). HRMS-ES (*m/z*): [M + Na]^+^: Calcd. for C_19_H_26_O_6_Na: 373.1627; found 373.1635.

Compound **5**: Yellow solid, m.p.: 98–100 °C. [α]${}_{D}^{20}$: +121.5 (*c* 9.1; CHCl_3_). IR (KBr; υ_max_: cm^−1^): 2958, 2927, 1747, 1466, 1379, 1250, 1169, 1086. ^1^H-NMR (δ = ppm, 200M Hz): 1.31–1.41 (1H, m, H-1); 1.52–1.69 (1H, m, H-2a); 2.17 (1H, ddd, *J* = 4, 8 and 12 Hz, H-2b); 4.07 (1H, ddd, *J* = 4 and 7 Hz, H-3); 3.42 (1H, d, *J* = 7 Hz, H-4); 1.52–1.69 (1H, m, H-6a); 1.73–1.88 (1H, m, H-6b); 3.10 (1H, m, H-7); 4.67 (1H, dtd, *J* = 8 and 11 Hz, H-8); 1.73–1.88 (1H, m, H-9a); 1.90–2.08 (1H, m, H-9b); 1.90–2.08 (1H, m, H-10); 5.53 (1H, d, *J* = 2 Hz, H-13); 6.26 (1H, d, *J* = 2 Hz, H-13′); 1.04 (3H, d, *J* = 7 Hz, H-14); 0.99 (3H, s, H-15); 0.13* (9H, s, H-1′, H-2′ and H-3′); 0.17*(9H, s, H-1″, H-2″ and H-3″). ^13^C-NMR (δ = ppm, 50M Hz): 41.1 (C-1); 36.8 (C-2); 69.7 (C-3); 78.1 (C-4); 44.6 (C-5); 40.4 (C-6); 37.7 (C-7); 79.9 (C-8); 36.5 (C-9); 30.3 (C-10); 140.2 (C-11); 169.8 (C-12); 122.3 (C-13); 16.6 (C-14); 18.5 (C-15); 0.40* (C-1′, C-2′ and C-3′); 0.50* (C-1″, C-2″ and C-3″). HRMS-ES (*m/z*): [M + Na]^+^: Calcd. for C_21_H_38_O_4_Si_2_Na: 433.2206; found 433.2209.

Compound **6**: Yellow solid, m.p.: 50–52 °C. [α]${}_{D}^{20}$: +104.41 (*c* 9.97; CHCl_3_). IR (KBr; *υ_max_*: cm^−1^): 2960, 2899, 1747, 1464, 1385, 1248, 1173, 1088. ^1^H-NMR (δ = ppm, 200M Hz): 1.31–1.39 (1H, m, H-1); 1.63–1.77 (1H, m, H-2a); 1.88–2.06 (1H, m, H-2b); 4.06 (1H, cd, J = 7 Hz, H-3); 3.4 (1H, d, J = 7 Hz, H-4); 1.52–1.57 (m, H6a); 1.76–1.88 (m, H6b); 3.09 (m, H7); 4.65 (ddd, J = 8 and 11 Hz, H8); 1.76–1.88 (m, H9a); 2.09–2.23(1H, m, H-9b); 1.88–2.06 (1H, m, H-10); 5.51 (1H, d, J = 3 Hz, H-13); 6.24 (1H, d, J = 3 Hz, H-13′); 1.02 (3H, d, J = 5 Hz, H-14); 0.98 (3H, s, H-15); -0,01(6H, s, H-1′ and H-2′); 0.61* (3H, d, J = 8 Hz, H-4′); 0.63* (3H, d, J = 8 Hz, H-5′); -0,01(6H, s, H-1″ and H-2″); 0.61* (3H, d, J = 8 Hz, H-4″); 0.63* (3H, d, J = 8 Hz, H-5″). ^13^C-NMR (δ = ppm, 50M Hz): 41.2 (C-1); 36.8 (C-2); 69.7 (C-3); 78.2 (C-4); 44.8 (C-5); 40.4 (C-6); 37.8 (C-7); 80.0 (C-8); 36.5 (C-9); 30.3 (C-10); 140.3 (C-11); 169.8 (C-12); 122.3 (C-13); 16.5 (C-14); 18.5 (C-15); -0,01 (C-1′, 2′ and 3′); 17.1 (C-4′); 17.0 (C-5′); −0,01 (C-1″, 2″ and 3″); 17.1 (C-4″); 17.0 (C-5″). HRMS-ES (*m/z*): [M + Na]^+^: Calcd. for C_25_H_46_O_4_Si_2_Na: 489.2832; found 489.2825.

Compound **7**: Yellow solid, m.p.: 70–72 °C. [α]${}_{D}^{20}$: +42.28 (*c* 10.36; CHCl_3_). IR (KBr; *υ_max_*: cm^−1^): 3537, 2962, 1763, 1427, 1267, 1113, 702. ^1^H-NMR (δ = ppm, 200M Hz): 1.23–1.29 (1H, m, H-1); 1.50–1.62 (1H, m, H-2a); 1.50–1.62 (1H, m, H-2b); 4.25 (1H, dd, *J* = 8 Hz, H-3); 3.45 (1H, dd, *J* = 8 Hz, H-4); 1.50–1.62 (1H, m, H-6a); 1.93–2.04 (1H, m, H-6b); 3.09 (1H, m, H-7); 4.58 (1H, ddd, *J* = 8 and 11 Hz, H-8); 1.70–1.79 (1H, m, H-9a); 2.06–2.18 (1H, m, H-9b); 1.80–1.91 (1H, m, H-10); 5.58 (1H, d, *J* = 2 Hz, H-13); 6.24 (1H, d, *J* = 2 Hz, H-13′); 0.95 (3H, d, *J* = 7 Hz, H-14); 1.05 (3H, s, H-15); 1.11 (9H, s, H-2′, H-3′ and H-4′); 7.63 (8H, m, H-2″ and H-3″); 7.43 (2H, m, H-4″). ^13^C-NMR (δ = ppm, 50M Hz): 41.1 (C-1); 35.8 (C-2); 70.3 (C-3); 77.0 (C-4); 44.3 (C-5); 40.6 (C-6); 37.6 (C-7); 79.7 (C-8); 36.4 (C-9); 30.0 (C-10); 139.8 (C-11); 169.9 (C-12); 122.8 (C-13); 16.6 (C-14); 17.8 (C-15); 0.2 (C-1′); 27.2 (C-2′); 133.4 (C-1″); 127.8 (C-2″); 135.9 (C-3″); 130.1 (C-4″). HRMS-ES (*m/z*): [M + Na]^+^: Calcd. for C_31_H_40_O_4_Si_2_Na: 527.2594; found 527.2586.

Compound **8**: White solid, m.p.: 100–101 °C. [α]${}_{D}^{20}$ +30.3 (*c* 4.84; CHCl_3_). IR (KBr; *υ_max_*: cm^−1^): 3523, 3278, 2918, 2114, 1761, 1385, 1269, 1120, 1086, 997. ^1^H-NMR (δ = ppm, 600M Hz): 1.45 (1H, m, H-1); 1.75–1.82 (1H, m, H-2a); 2.05–2.12 (1H, m, H-2b); 4.26 (1H, m, H-3); 3.47 (1H, d, *J* = 7 Hz, H-4); 1.60 (1H, t, *J* = 1, 4 and 7 Hz, H-6a); 1.95–2.01 (1H, m, H-6b); 3.16 (1H, m, H-7); 4.68 (1H, ddd, *J* = 8 and 11 Hz, H-8); 1.72–1.81 (1H, m, H-9a); 2.15–2.20 (1H, m, H-9b); 1.95–2.01 (1H, m, H-10); 5.55 (1H, s, H-13); 6.26 (1H, d, J = 2 Hz, H-13′); 1.04 (3H, d, *J* = 7 Hz, H-14); 1.02 (3H, s, H-15); 4.36 (1H, d, *J* = 2 Hz, H-1′a); 4.33 (1H, d, *J* = 2 Hz, H-1′b); 2.5 (1H, s, H-3′). ^13^C-NMR (δ = ppm, 150M Hz): 41.6 (C-1); 35.1 (C-2); 66.5 (C-3); 83.9 (C-4); 44.2 (C-5); 40.7 (C-6); 38.0 (C-7); 79.9 (C-8); 36.4 (C-9); 30.2 (C-10); 140.2 (C-11); 169.8 (C-12); 122.7 (C-13); 16.6 (C-14); 18.8 (C-15); 58.5 (C-1′); 75.2 (C-2′); 80.0 (C-3′). HRMS-ES (*m/z*): [M + Na]^+^: Calcd. for C_18_H_24_O_4_Na: 327.1572; found 327.1574.

Compound **9**: White solid, m.p.: 125–126 °C. [α]${}_{D}^{20}$: +59.0 (*c* 5.73; CHCl_3_). IR (KBr; *υ_max_*: cm^−1^): 3431, 2927, 2355, 2119, 1759, 1387, 1273, 1088, 997. ^1^H-NMR (δ = ppm, 600M Hz): 1.48 (1H, m, H-1); 1.74–1.85 (1H, m, H-2a); 2.055–2.12 (1H, m, H-2b); 4.25 (1H, m, H-3); 3.62 (1H, d, *J* = 7 Hz, H-4); 1.60 (1H, t, *J* = 15 Hz, H-6a); 1.97–2.04 (1H, m, H-6b); 3.18 (1H, m, H-7); 4.69 (1H, ddd, *J* = 8 and 11 Hz, H-8); 1.74–1.85 (1H, m, H-9a); 2.16–2.22 (1H, m, H-9b); 1.97–2.04 (1H, m, H-10); 5.58 (1H, s, H-13); 6.26 (1H, s, H-13′); 1.05 (3H, d, *J* = 7 Hz, H-14); 1.04 (3H, s, H-15); 4.43 (1H, d, *J* = 16 Hz, H-1′a); 4.30 (1H, d, *J* = 1 6 Hz, H-1′b); 2.46 (1H, brs, H-3′); 4.22 (1H, d, J = 16 Hz, H-1″a); 4.16 (1H, d, J = 16 Hz, H-1″b); 2.42 (1H, brs, H-3″). ^13^C-NMR (δ = ppm, 150M Hz): 41.4 (C-1); 32.9 (C-2); 72.2 (C-3); 81.7 (C-4); 44.0 (C-5); 40.1 (C-6); 37.8 (C-7); 79.8 (C-8); 36.4 (C-9); 30.2 (C-10); 140.0 (C-11); 169.7 (C-12); 122.6 (C-13); 16.6 (C-14); 18.4 (C-15); (57.0C-1′); 80.2 (C-2′); 74.5 (C-3′); 56.4 (C-1″); 79.7 (C-2″); 74.3 (C-3″) HRMS-ES (*m/z*): [M + Na]^+^: Calcd. for C_21_H_26_O_4_Na: 365.1729; found 365.1729.

Compound **10**: White amorphous solid, m.p.: 74–75 °C. [α]${}_{D}^{20}$: +10.5 (*c* 3.80; CHCl_3_). IR (KBr; *υ_max_*: cm^−1^): 3433, 2924, 2360, 1757, 1637, 1456, 1205, 1076, 725. ^1^H-NMR (δ = ppm, 600M Hz): 1.41–1.45 (1H, m, H-1); 1.72–1.79 (1H, m, H-2a); 2.07–2.11 (1H, m, H-2b); 4.15 (1H, m, H-3); 3.52 (1H, d, *J* = 7 Hz, H-4); 1.51 (1H, t, *J* = 15 Hz, H-6a); 1.68 (1H, dd, *J* = 15 Hz, H-6b); 3.07 (1H, m, H-7); 4.64 (1H, ddd, *J* = 3 and 8- Hz, H8); 1.72–1.79 (1H, m, H-9a); 2.13–2.19 (1H, m, H-9b); 1.94–1.99 (1H, m, H-10); 5.43 (1H, d, *J* = 2 Hz, H-13); 6.2 (1H, d, *J* = 2 Hz, H-13′); 1.03 (3H, d, J = 7 Hz, H-14); 1.00 (3H, s, H-15); 4.75 (2H, d, *J* = 12, H-1′); 7.53 (1H, s, H-3′); 5.56 (2H, d, *J* = 3, H-1″); 7.40 (2H, m, H-3″ and H-5″); 7.30 (1H, m, H-4″). ^13^C-NMR (δ = ppm, 150M Hz): 41.3 (C-1); 34.9 (C-2); 66.3 (C-3); 84.4 (C-4); 44.2 (C-5); 40.7 (C-6); 37.8 (C-7); 79.8 (C-8); 36.3 (C-9); 30.0 (C-10); 139.8 (C-11); 169.7 (C-12); 122.7 (C-13); 16.6 (C-14); 18.7 (C-15); 63.8 (C-1′); 145.1 (C-2′); 122.7 (C-3′); 54.4 (C-1″); 134.2 (C-2″); 129.3 (C-3″); 128.2 (C-4″); 129.0 (C-5″). HRMS-ES (*m/z*): [M + Na]^+^: Calcd. for C_25_H_31_N_3_O_4_Na: 460.2212; found 460.2217.

Compound **11**: White amorphous solid, m.p.: 80–81 °C. [α]${}_{D}^{20}$: +7.3 (*c* 1.79; CHCl_3_). IR (KBr; *υ_max_*: cm^−1^): 3433, 2962, 2924, 1757, 1456, 1261, 1092, 1120, 800. ^1^H-NMR (δ = ppm, 600M Hz): 1.37–1.42 (1H, m, H-1); 1.71–1.77 (1H, m, H-2a); 2.01–2.04 (1H, m, H-2b); 3.96–4.00 (1H, m, H-3); 3.45 (1H, d, *J* = 7 Hz, H-4); 1.68 (1H, dd, *J* = 15 Hz, H-6a); 1.71–1.77 (1H, m, H-6b); 3.03 (1H, m, H-7); 4.61–4.65 (1H, m, H-8); 1.71–1.77 (1H, m, H-9a); 2.13–2.18 (1H, m, H-9b); 1.95–1.98 (1H, m, H-10); 5.43 (1H, d, *J* = 2 Hz, H-13); 6.19 (1H, d, *J* = 2 Hz, H-13′); 1.01 (3H, d, *J* = 7 Hz, H-14); 0.99 (3H, s, H-15); 4.64–4.67 (2H, m, H-1′a) / 4.65–4.71 (2H, m, H-1′a); 4.47 (2H, d, *J* = 12, H-1′b) / 4.54 (2H, d, *J* = 12, H-1′b); 7.61 (2H, s, H-3′a); 7.58 (1H, s, H-3′b); 5.58 (d, J = 10, H-1″a); 5.54 (d, J = 10, H-1″b); 7.29–7.40 (10H, m, H-3″, H-4″ and H-5″). ^13^C-NMR (δ = ppm, 150M Hz): 41.0 (C-1); 33.2 (C-2); 73.9 (C-3); 83.6 (C-4); 44.3 (C-5); 40.3 (C-6); 37.8 (C-7); 79.8 (C-8); 36.3 (C-9); 30.1 (C-10); 139.7 (C-11); 169.8 (C-12); 122.9 (C-13); 16.7 (C-14); 18.5 (C-15); 62,3 (C-1′a); 63.2 (C-1′b); 145.2 (C-2′a); 145.3 (C-2′b); 123.1 (C-3′a); 123.4 (C-3′b); 54.5 (C-1″); 134.4 (C-2″); 128.3 (C-3″); 129.2 (C-4″); 129.0 (C-5″). HRMS-ES (*m/z*): [M + Na]^+^: Calcd. for C_35_H_40_N_6_O_4_Na: 631.3009; found 631.3015.

Compound **12:** Amorphous solid, m.p.: 164–165 °C. [α]${}_{D}^{20}$: −76.3 (*c* 9.3; CHCl_3_). IR (KBr; *υ_max_*: cm^−1^): 2927, 2854, 1767, 1734, 1709, 1657, 1466, 1238, 949. ^1^H-NMR (δ = ppm, 200M Hz): 3.02 (1H, dt, *J* = 3 and 2 Hz, H-1); 7.65 (1H, dd, *J* = 2 and 6 Hz, H-2); 6.07 (1H, dd, *J* = 3 and 6 Hz, H-3); 5.38 (1H, br s, H-6); 3.51 (1H, ddd, *J* = 8 and 12, H-7); 4.88 (1H, ddd, *J* = 2 and 7 Hz, H-8); 1.75 (1H, cd, *J* = 19 Hz, H-9a); 2.43 (1H, m, H-9b); 2.07–2.18 (1H, m, H-10); 6.14 (1H, d, *J* = 3 Hz, H-13); 6.45 (1H, d, J = 3 Hz, H-13′); 1.27 (3H, d, *J* = 7 Hz, H-14); 1.01 (3H, s, H-15); 2.00 (3H, brs, H-2′). ^13^C-NMR (δ = ppm, 50M Hz): 53.2 (C-1); 162.1 (C-2); 129.7 (C-3); 208.9 (C-4); 55.4 (C-5); 77.6 (C-6); 47.7 (C-7); 78.1 (C-8); 40.2 (C-9); 26.1 (C-10); 137.4 (C-11); 169.7 (C-12); 125.0 (C-13); 20.0 (C-14); 18.4 (C-15); 169.5 (C-1′); 21.0 (C-2′). HRMS-ES (*m/z*): [M + Na]^+^: Calcd. for C_17_H_20_O_5_Na: 327.1208; found 327.1215.

Compound **13:** Amorphous solid. m.p.: 82–84 °C. [α]${}_{D}^{20}$: −41.6 (*c* 11.2; CHCl_3_). IR (KBr; *υ_max_*: cm^−1^): 2960, 2929, 1761, 1714, 1464, 1246, 1109, 845. ^1^H-NMR (δ = ppm, 200M Hz): 3.20 (1H, dt, *J* = 6 and 12 Hz, H-1); 7.64 (1H, dd, *J* = 2 and 6 Hz, H-2); 5.99 (1H, dd, *J* = 3 and 6 Hz, H-3); 4.42 (1H, d, *J* = 2 Hz, H-6); 3.30 (1H, m, H-7); 4.95 (1H, td, *J* = 3 and 8 Hz, H-8); 1.78 (1H, cd, *J* = 8 Hz, H-9a); 2.30 (1H, m, H-9b); 2.22–2.33 (1H, m, H-10); 5.76 (1H, d, *J* = 3 Hz, H-13); 6.36 (1H, d, J = 3 Hz, H1–3′); 1.24 (3H, d, *J* = 7 Hz, H-14); 0.9 (3H, s, H-15); 0.11 (9H, s, H-1′, H-2′ and H-3′). ^13^C-NMR (δ = ppm, 50M Hz): 51.5 (C-1); 163.4 (C-2); 129.2 (C-3); 210.8 (C-4); 57.6 (C-5); 76.0 (C-6); 51.7 (C-7); 78.7 (C-8); 40.1 (C-9); 26.0 (C-10); 138.1 (C-11); 170.0 (C-12); 122.9 (C-13); 20.1 (C-14); 18.4 (C-15); 0.4 (C-1′, 2′ and 3′). HRMS-ES (*m/z*): [M + Na]^+^: Calcd. for C_18_H_26_O_4_SiNa: 357.1498; found 357.1492.

Compound **14:** Amorphous solid, m.p.: 50–52 °C. [α]${}_{D}^{20}$: −33.9 (*c* 9.85; CHCl_3_). IR (KBr; *υ_max_*: cm^−1^): 2958, 2922, 2364, 1770, 1718, 1464, 1269, 1090, 858. ^1^H-NMR (δ = ppm, 200M Hz): 3.21 (1H, dt, *J* = 3 and 2 Hz, H-1); 7.64 (1H, dd, *J* = 2 and 6 Hz, H-2); 5.98 (1H, dd, *J* = 3 and 6 Hz, H-3); 4.42 (1H, d, *J* = 2 Hz, H-6); 3.39 (1H, m, H-7); 4.92 (1H, td, *J* = 3 and 8 Hz, H-8); 1.76 (1H, cd, *J* = 8 Hz, H-9a); 2.33 (1H, m, H-9b); 2.1 (1H, m, H-10); 5.74 (1H, d, *J* = 3 Hz, H-13); 6.36 (1H, d, *J* = 3 Hz, H-13′); 1.23 (3H, d, *J* = 7 Hz, H-14); 0.88 (3H, s, H-15); 0.12*(3H, s, H-1′); 0.44*(3H, s, H-2′); 0.94 (1H, s, H-3′); 0.84*(3H, s, H-4′); 0.90*(3H, s, H-5′). ^13^C-NMR (δ = ppm, 50M Hz): 51.8 (C-1); 163.2 (C-2; 129.2 (C-3); 210.7 (C-4); 57.8 (C-5); 76.1 (C-6); 51.5 (C-7); 78.7 (C-8); 40.3 (C-9); 26.0 (C-10); 138.1 (C-11); 170.0 (C-12); 123.0 (C-13); 20.1 (C-14); 18.5 (C-15); −3.6* (C-1′); -3.8* (C-2′); 29.7 (C-3′); 16.8 (C-4′ and 5′). HRMS-ES (*m/z*): [M + Na]^+^: Calcd. for C_20_H_30_O_4_SiNa: 385.1811; found 385.1819.

Compound **15**: Amorphous solid, m.p.: 150–151 °C. [α]${}_{D}^{20}$: −94.3 (*c* 5.23; CHCl_3_). IR (KBr; *υ_max_*: cm^−1^): 2960, 2927, 1763, 1714, 1468, 1379, 1053, 847. ^1^H-NMR (δ = ppm, 200M Hz): 7.52 (1H, d, *J* = 6 Hz, H-2); 6.19 (1H, d, *J* = 6 Hz, H-3); 4.83 (1H, d, *J* = 9 Hz, H-6); 3.23 (1H, m, H-7); 1.92–2.05 (1H, m, H-8); 1.62–1.69 (1H, m, H-9); 1.62–1.69 (1H, m, H-10); 5.50 (1H, d, *J* = 3 Hz, H-13); 6.26 (1H, d, *J* = 4 Hz, H-13′); 1.05 (3H, d, *J* = 5 Hz, H-14); 1.03 (3H, s, H-15); 0.15 (9H, brs, H-1′, H-2′ and H-3′). ^13^C- NMR (δ = ppm, 50M Hz): 86.5 (C-1); 163.0 (C-2); 131.0 (C-3); 207.6 (C-4); 58.2 (C-5); 79.7 (C-6); 41.8 (C-7); 25.6 (C-8); 32.2 (C-9); 38.9 (C-10); 170.0 (C-11); 139.1 (C-12); 120.3 (C-13); 17.7 (C-14); 15.4 (C-15); 2.6 (C-1′, 2′ and 3′). HRMS-ES (*m/z*): [M + Na]^+^: Calcd. for C_18_H_26_O_4_SiNa: 357.1498; found 357.1502.

### 3.6. Cells, Culture, and Plating

The following human solid tumor cell lines were used in this study: HBL-100 (breast), HeLa (cervix), SW1573 (non-small cell lung), T-47D (breast), A549 (lung), and WiDr (colon). Cells were maintained in 25 cm^2^ culture flasks in RPMI 1640 supplemented with 5% heat-inactivated fetal calf serum and 2 mM L-glutamine in a 37 °C, 5% CO_2_, 95% humidified air incubator. Exponentially growing cells were trypsinized and resuspended in antibiotic containing medium (100 units penicillin G and 0.1 mg of streptomycin per mL). Single-cell suspensions displaying >97% viability by trypan blue dye exclusion test were subsequently counted. After counting, dilutions were made to give the appropriate cell densities for inoculation onto 96-well microtiter plates. Cells were inoculated in a volume of 100 μL per well at densities of 20,000 (WiDr), 15,000 (T-47D and A549), and 10,000 (HeLa, SW1573, and HBL-100) cells per well, based on their doubling times.

### 3.7. Antiproliferative Tests

Chemosensitivity tests were performed using the SRB assay of the NCI with slight modifications. Briefly, pure compounds were initially dissolved in DMSO at 400 times the desired final maximum test concentration. Control cells were exposed to an equivalent concentration of DMSO (0.25% *v*/*v*, negative control). Each agent was tested in triplicate at different dilutions in the range 1–100 μM. Drug treatment started on day 1 after plating. Drug incubation periods were 48 h, after which cells were precipitated with 25 μL of ice-cold 50% (*w*/*v*) trichloroacetic acid and fixed for 60 min at 4 °C. Then, the SRB assay was performed. Optical density (OD) of each well was measured at 492 nm, using BioTek’s PowerWave XS Absorbance Microplate Reader (Highland Park, Winooski, VT, USA). Values were corrected for background OD from wells containing only culture medium. The percentage growth (PG) was calculated with respect to untreated control cells (C) at each level of drug concentrations based on the difference in OD at the start time (T0) and at the end of drug exposure (T), according to NCI formulas. Therefore, if T is greater than or equal to T0, the calculation is 100 × [(T–T0)/(C–T0)]. If T is lower than T0, denoting cell death, the calculation is 100 × [(T–T0)/(T0)]. The effect is defined as the growth percentage, where 50% growth inhibition (GI_50_) represents the concentration at which PG is +50. Based on these calculations, a PG value of 0 corresponds to the number of cells present at the beginning of drug exposure, while negative PG values denote net cell death.

### 3.8. Cytotoxicity on Primary Cell Culture

In a 96-well plate, spleen cells from Balb/c mouse (1.5 × 10^5^) were incubated with different drug dilutions (200, 100, 50, 10, and 5 μg/mL) in RPMI medium containing 10% fetal calf serum. After 48 h of incubation at 37 °C (5% CO_2_), cells were harvested, washed once with PBS, and stained with 2.5 μg/mL propidium iodide (PI) for 5 min at room temperature. Subsequently, cell death was assessed by flow cytometry using a BD FACSaria II cytometer. Cells incubated only with a drug vehicle were used as a 100% viability control and death percentage was calculated according to the following formula:$$Death\left(\%\right)=\left[1-\frac{{\left(\%PI^{-}cells\right)}_{drug-treated}}{{\left(\%PI^{-}cells\right)}_{100\%\;viability\;control}}\right]\times 100.$$

Then the concentration capable of causing cell death in 50% of splenocytes (CC_50_) was determined using a non-linear regression approach.

### 3.9. Statistical Analysis

Results are presented as means ± SD. GraphPad Prism 5.0 software (GraphPad Software Inc., San Diego, CA, USA) was employed to carry out calculations. The results account for three to four independent experiments.

## 4. Conclusions

Given these results, using natural compounds can be a viable strategy to prepare new active molecules. Sesquiterpene lactones were the starting material for their transformation into several oxygenated and oxo-nitrogenated derivatives by chemical reactions aiming at the hydroxylated positions. Our strategy was to obtain new derivatives, including functionalities such as acetates, silyl ethers, and 1,2,3-triazoles. Although sesquiterpene lactones showed interesting antiproliferative activity values, a significant number of these synthetic derivatives showed greater activity than the naturally occurring parent product. Many of the synthesized analogs were more selective toward tumor cell lines than normal cells. Compound **11**, the ditriazolyl cumanin derivative, proved to be more active and selective than cumanin in the tested breast, cervix, lung, and colon tumor cell lines. Thus, this compound can be considered for further studies and is a possible candidate for developing new antitumor agents.

Finally, this work aims at illustrating the possibility of obtaining new naturally-occurring anti-tumor leads from molecular frameworks, some of which exhibit significantly improved bioactivity just by common chemical transformations.

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
