# Peer review of "Preparation of Sesquiterpene Lactone Derivatives: Cytotoxic Activity and Selectivity of Action"

_molecules, 2019, doi:10.3390/molecules24061113_

Round 1

Reviewer 1 Report

In the manuscript, the synthesis of a series of derivatives of cumanin, helenalin and hymenin were semi-synthesized through acetylation, silyl etherification, or introducing triazole moiety. The cytotoxic effect of the synthesized compounds also evaluated. The research work of this paper mainly lies in the research of discovery of new anti-tumor compounds. The manuscript can be considered for publication after minor revisions. Some detail points were listed as below,

1.       It would be better to change the title of the manuscript. “oxygenated and oxonitrogenated derivatives” was not suitable since all of the derivatives were from the etherification or esterification of hydroxyl groups.

2.       The introduction fraction should be rewritten. To delet the general knowledge content (for example, paragraph 1 and 2).

3.       The HMBC, HSQC and COSY spectra of the mono-substituent derivatives 7 and 8 should be determined and analyzed to confirm the position of substituent. 

Author Response

Thanks for your comments and suggestions. Our responses are shown below. Modifications have been included in the manuscript.

Response Point 1:

The title was changed to "Preparation of sesquiterpene lactone derivatives. Cytotoxic activity and selectivity of action".

Response Point 2:

The introduction has been improved according to the request.

Response Point 3:

The position of the substituent at C3, compound 7, has been described on page 3, lines 91-100 (new version). The COSY, HMBC and HSQC spectra were included in the supplementary material section as supporting material. In the case of compound 8, the position of the substituent at C3 was made based on the observations seen for compound 7. The COSY, HMBC and HSQC spectra are included in the supplementary material section for supporting the assignments that were made.

Reviewer 2 Report

The manuscript present the results on synthesis of oxygenated and oxy-nitrogenated derivatives of sesquiterpene lactones. Next, the antiproliferative and cytotoxic activities were evaluated and compared with the activities of parent compounds. Moreover, the selectivity index was determined.

However, it is not certain if the differences between of antiproliferative activity demonstrated by newly synthesised and parent compounds are statistically valid.

Why did not Authors used MTT to asses cytotoxicity of investigated compounds? The IC50 value could be calculated. The MTT test is the most popular reference test and the results could be more easily compared with previously published.

The Latin names of plant species should be given in Italic font.

Author Response

Thanks for your comments and suggestions. Our responses are shown below. Modifications have been included in the manuscript.

Response Point 1:

The statistical differences between the antiproliferative activity (GI50 values) demonstrated by synthesized and parent compounds has been included in Table 1 of the manuscript. We have also included the statistical differences between the cytototoxic activity on normal cells (CC50 values) of the derivatives and each natural compound (Table 2).

Response Point 2:

The most well-known screening programme worldwide for the discovery of new antitumor drugs is the one of the US National Cancer Institute's NCI-60 (https://dtp.cancer.gov/discovery_development/nci-60/default.htm) [1]. This programme uses the sulforhodamine B (SRB) assay to calculate the effects of drugs on cell growth. The equivalence with the MTT test has been demonstrated long time ago [2] and the advantages of SRB over MTT have been disclosed [3].

The NCI renamed the IC50 value (the concentration that causes 50% growth inhibition) by the GI50 value to emphasize the correction for the cell count at time zero [2].

[1] Chabner BA. NCI-60 Cell Line Screening: A Radical Departure in its Time. J Natl Cancer Inst. 2016, 108(5). pii: djv388. doi: 10.1093/jnci/djv388.

[2] Monks A et al. Feasibility of a high-flux anticancer drug screen using a diverse panel of cultured human tumor cell lines. J Natl Cancer Inst. 1991, 83(11):757-66.

[3] Keepers Y, et al. Comparison of the sulforhodamine B protein and tetrazolium (MTT) assays for in vitro chemosensitivity testing. Eur J Cancer 1991, 27(7):897-900.

Response Point 3:

This point was corrected in the revised version of the manuscript

Reviewer 3 Report

The justification for this study is lacking, I dont understand why putting silyl protecting groups (which make the compound even less drug like and dramatically increase lipophilicity) woud be a good idea in the search for antitumor compounds. Furthermore the reason or discussion behind the appendage of a benzyl triazole is not there. In principal at least this could be worthwhile if a series of structurally and electronically diverse (H-bond acceptor/donors, hydrophilic, hydrophobic, pi-stacking etc) triazoles were synthesised and a SAR established. In its current form this paper does nothing to advance science.

Author Response

Thanks for your comments and suggestions. Our responses are shown below. Modifications have been included in the manuscript.

Response:

The introduction of silicon groups is directly related to the increase in lipophilicity. In fact, the increase in the activity of compounds 5, 6, 7, 13 and 14 and selectivity has been demonstrated. This justification is based on experimental data, which have been reported previously:

1-         The tert-butyl dimethyl silyl group as an enhancer of drug cytotoxicity against human tumor cells. Osvaldo J. Donadel, Tomás Martín, Víctor S. Martín, Jesús Villar and José M. Padrón. Bioorg. Med. Chem. Lett.,2005, 15, 3536–3539. ISSN: 0960-894X.

2-         Enhancement of drug cytotoxicity by silicon containing groups. José M. Padrón, Osvaldo J. Donadel, Leticia G. León, Tomás Martín and Víctor S. Martín. Letter drug design and discovery, 2006, 3(1), 625-630. ISSN: 1570-1808.

Round 2

Reviewer 3 Report

Increasing lipophilicity may well result in an increase in cytotoxicity in vitro. However, this is a long way from being clinically relevant, I recommend accepting the article however I think it is important to acknowledge the limitations of this strategy in the context of drug design.